# Hedgehog and Gpr161: Regulating cAMP Signaling in the Primary Cilium

**DOI:** 10.3390/cells9010118

**Published:** 2020-01-03

**Authors:** Philipp Tschaikner, Florian Enzler, Omar Torres-Quesada, Pia Aanstad, Eduard Stefan

**Affiliations:** 1Institute of Biochemistry and Center for Molecular Biosciences, University of Innsbruck, Innrain 80/82, 6020 Innsbruck, Austria; philipp.tschaikner@uibk.ac.at (P.T.); florian.enzler@uibk.ac.at (F.E.); omar.quesada@uibk.ac.at (O.T.-Q.); 2Institute of Molecular Biology, University of Innsbruck, 6020 Innsbruck, Austria; pia.aanstad@uibk.ac.at

**Keywords:** kinase, anchoring proteins, AKAP, orphan GPCR, signalosome, second messenger, cAMP, adenylyl cyclase, Hedgehog signaling, molecular interactions, dynamic complexes, signaling nodes, signaling circuits, compartmentalization

## Abstract

Compartmentalization of diverse types of signaling molecules contributes to the precise coordination of signal propagation. The primary cilium fulfills this function by acting as a spatiotemporally confined sensory signaling platform. For the integrity of ciliary signaling, it is mandatory that the ciliary signaling pathways are constantly attuned by alterations in both oscillating small molecules and the presence or absence of their sensor/effector proteins. In this context, ciliary G protein-coupled receptor (GPCR) pathways participate in coordinating the mobilization of the diffusible second messenger molecule 3′,5′-cyclic adenosine monophosphate (cAMP). cAMP fluxes in the cilium are primarily sensed by protein kinase A (PKA) complexes, which are essential for the basal repression of Hedgehog (Hh) signaling. Here, we describe the dynamic properties of underlying signaling circuits, as well as strategies for second messenger compartmentalization. As an example, we summarize how receptor-guided cAMP-effector pathways control the off state of Hh signaling. We discuss the evidence that a macromolecular, ciliary-localized signaling complex, composed of the orphan GPCR Gpr161 and type I PKA holoenzymes, is involved in antagonizing Hh functions. Finally, we outline how ciliary cAMP-linked receptor pathways and cAMP-sensing signalosomes may become targets for more efficient combinatory therapy approaches to counteract dysregulation of Hh signaling.

## 1. Introduction

Compartmentalization is an important aspect of intracellular signaling, both in contributing to the precise coordination of signal propagation and in ensuring specificity of signaling outcomes [1,2]. Primary cilia are microtubule-based antenna-like structures that extend from the plasma membrane, and represent a highly specialized subcellular compartment/environment for sensing and signaling [3,4]. The composition of the cilium, both in terms of signaling molecules, and also the lipid composition of the ciliary plasma membrane, is strictly regulated, with the transition zone operating to restrict diffusion and control trafficking in and out of the cilium [5]. The last couple of decades have seen an increased focus on, and understanding of, the role that cilia play in regulating signaling. This led to the identification of several signaling molecules, including G-protein coupled transmembrane receptors (GPCRs) and second messengers, that localize to cilia [3,6,7].

One central ciliary signaling component is the conserved and ubiquitous second messenger 3′, 5′-cyclic adenosine monophosphate (cAMP). First discovered by Sutherland and Rall in 1958, cAMP was proposed to act as a signal effector molecule downstream of hormones like epinephrine, dopamine, and prostaglandin [8]. Although this proposal was initially met with skepticism because it was difficult to see how the various actions of distinct hormones could all be mediated by a single effector molecule, we now know that second messengers like cAMP direct different cellular outcomes by interacting with different binding partners in distinct subcellular locations. Through cAMP-dependent protein kinase A (PKA), one of its primary targets, cAMP also plays a pivotal role in the regulation of Hedgehog (Hh) signaling in primary cilia [5,9,10]. This review will focus on the regulation and compartmentalization of cAMP in the primary cilium in the context of Hh signaling. We will first briefly describe the main effector molecules of cAMP, and how oscillations of cAMP fluxes are coordinated. Second, we discuss the function of PKA, the prototypical cAMP-effector, and one of the best studied examples for protein allostery and second messenger/protein interactions [11]. We discuss the role of spatiotemporal organization in receptor controlled kinase signaling by highlighting PKA signalosomes in, and at the base of, the primary cilium. Third, we describe the ways that Hh and the orphan receptor Gpr161 interact to regulate PKA activity and Hh signaling outcomes. In this context, we finally present an outlook of how these antagonizing signaling pathways in the primary cilium may become targets for combinatory pharmaceutical interference.

## 2. cAMP Mobilization

Intracellular cAMP mobilization is regulated by a family of adenylyl cyclases (ACs), which synthesize cAMP from adenosine triphosphate (ATP) (Figure 1) [12]. The activity of ACs is controlled by numerous extracellular cues, and mediates defined receptor–effector interactions. Prominent examples of extracellular stimuli and hormones activating ACs are epinephrine, dopamine, prostaglandin E2, and glucagon, which bind to defined cell surface receptors of the GPCR superfamily. GPCRs represent the largest family of cell surface molecules involved in signal transmission, and control a plethora of different cellular functions [13,14,15]. Here, we discuss GPCR coupling to the AC controlling G proteins Gαs and Gαi, which are part of the trimeric Gα/β/γ G protein complex. On the molecular level, agonist binding to specific GPCRs leads to the activation of the AC-stimulatory Gαs proteins [13,16,17,18,19,20,21]. In addition, AC-inhibitory GPCR pathways also exist (=Gαi protein coupling), which contribute to reducing AC activities, and thus cAMP synthesis [15,20,22]. ACs synthesize cAMP from ATP by enzyme activities that are, in most cases, located on the inner side of the plasma membrane. This promotes spatiotemporally restricted cAMP oscillations, which are regulated and fine-tuned in different ways. One mechanism involves receptor complex desensitization, for switching off receptor activation at the plasma membrane. However, it should be noted that internalized receptor–effector complexes can engage in signal propagation and that receptor recycling and synthesis is critical for reactivation of signal transmission and oscillations [14,23,24,25]. Another way to limit the diffusion or abundance of cAMP is achieved by the presence and activation of phosphodiesterases (PDEs) [26,27,28,29,30]. Different PDEs mediate the breakdown of cAMP by hydrolyzing it to AMP [31]. Using visualization technologies, the existence of cell- and context-specific cAMP nanodomains has been demonstrated [32,33,34,35,36,37,38,39,40]. Spatially separated pools of cAMP are likely to be engaged in different functional outputs. This is based on the fact that a collection of different cAMP binding proteins and complexes exist. These cAMP sensing signalosomes differ in their composition, which is displayed by the cell type-specific availability of different isoforms and varying macromolecular complexes [11,41,42,43,44,45].

## 3. cAMP Dynamics: Sensors and Effectors

Prime examples of cAMP-binding proteins are the functionally diverse type I and II holoenzyme complexes of PKA, isoforms of cAMP-dependent exchange proteins (Epac), cyclic nucleotide-gated ion channels (CNGs), and Popeye domain-containing proteins (POPDC) [34,35,46,47,48,49]. Before we discuss PKA, the prototype of an allosterically acting kinase complex and universal cAMP sensor, we briefly list the three other cAMP binding proteins. First, Epacs contain an evolutionally conserved cAMP-binding domain similar to PKA regulatory subunits (R). cAMP binding to Epac promotes activation of Rap, a small GTPase, which participates, among others, in the regulation of tumour cell invasion and metastasis via initiating of and/or interfering with ERK signaling [46,50,51,52]. Second, POPDC proteins contain functional cyclic nucleotide binding domains with cAMP binding affinities similar to PKA [53,54]. Mechanistic insights into cAMP binding functions are still elusive, but it is assumed that cAMP interactions with POPDC proteins trigger a conformational change and influences functional PPIs with, for example, TREK1-channels [48]. Third, cAMP binds to membrane localized CNGs to regulate their cation gating properties [49,55]. CNG functions are linked to signal transduction processes in olfactory receptor neurons [56]. In contrast to EPACs and POPDCs, which have not been directly linked to ciliary functions, CNGs localise to the membranes of specialized cilia of the olfactory sensory neurons, and thus are central for the translation of sensory cues into neuronal activity [57].

The prototypical and macromolecular cAMP-effector protein complex is nucleated by the tetrameric PKA holoenzyme [11,44]. Different compositions of these kinase holoenzymes mediate their functions in precise cell compartments and subcellular nanodomains. The core components are the PKA holoenzyme tetramers, which consist of two catalytic PKA subunits (C, PKAc), which interact with a dimer of identical regulatory PKA subunits (R). The cAMP-sensing R homodimer blocks C subunit activities by binding. Three genes for PKAc (coding for the proteins PKAc-α, PKAc-β, and PKAc-γ) and four genes for the R subunits (coding for the proteins RIα, RIβ, RIIα, and RIIβ) exist in the human genome [58,59,60,61]. The type I and II PKA R subunits are the nexus for cAMP-sensing and kinase activation, and are also responsible for the scaffold-mediated targeting of the holoenzyme to specific cAMP nanodomains. Distinct A kinase anchoring proteins (AKAPs) act as scaffolds to coordinate and ensure spatially and temporally restricted phosphorylation of substrates [2,41,42,43,45].

AKAPs sequester PKA holoenzymes to specific subcellular nanodomains. The structures of PKA type I and type II holoenzyme complexes differ considerably in their functions and solved structures [11,62,63,64,65,66]. Most AKAPs identified to date have been shown to be specific for type II holoenzymes. Moreover, it has been shown by several studies, using different kinase scaffolds, that type II PKA holoenzyme complex-specific AKAPs are often localized to subcellular organelles, including the nuclear envelope; the Golgi; and, for example, the cytoskeleton [41,42,67]. However, recently, two type I PKA selective AKAPs with low nM affinity for RI PKA subunits have been identified, the small AKAP smAKAP and the orphan GPCR Gpr161, respectively [45,68,69,70]. The function of the latter is discussed in the next section. However, the general principle of how AKAPs interact with PKA R subunit dimers is similar, based on a short interaction motif in the scaffold proteins. The conserved dimerization and docking domain (D/D domain) of R subunits dimerizes and forms an anti-parallel X-type helical bundle, thereby creating a hydrophobic surface for AKAP binding [71,72]. All AKAPs identified so far share a short amphipathic α-helix, which is composed of hydrophobic residues to diagonally dock the binding groove provided by the PKA R subunit dimer [41,42,43,68,69,73]. The majority of the so far more than 50 identified AKAPs have a modular organization and the capability to bind and compartmentalize combinations of kinase substrates, activator proteins, effector molecules, transduction, and termination enzymes. In Figure 2, we illustrate how AKAPs may function as broad signaling and scaffolding platforms for coordinating context-specific PPIs, as well as physical associations with PDEs, and the ubiquitin proteasome system (UPS) [29,74]. The AKAP-specific targeting domains recruit defined enzyme containing signalosomes to specific subcellular cell compartments. These actions are required to relay and restrict second messenger mediated signal transmission in a precise manner [41,42,43]. Compartmentalization of macromolecular kinase complexes to the plasma membrane, nucleus, mitochondria, endoplasmic reticulum, sarcoplasmic reticulum, primary cilium, or other distinct cellular organelles creates response specificity and guides the PKA-mediated information flow. Here, we would like to refer to reviews that describe the distinct subcellular localization of selected AKAPs and their interactions with, for example, additional kinases, phosphatases, PDEs, receptors, channels, ACs, Epacs, and other scaffolding proteins or factors relevant for GTPase involved signal propagation [2,35,41,42,43,67,75,76,77].

## 4. PKA and Hh in the Primary Cilium

Nearly every cell in the body assembles one primary non-motile cilium. The primary cilium is an antennae-like, microtubule-based protrusion from the apical membrane of most mammalian cell types [78]. Dynamics and cAMP activation patterns in the cilium are functionally interwoven with Hh signaling in the vertebrate cilia. The Hh pathway is evolutionarily conserved and plays essential roles in cell fate specification and cell proliferation during embryonic development and in adult tissue homeostasis [9,10,79]. PKA is a central negative regulator of Hh signaling, which, in the absence of Hh ligands (= pathway off state), phosphorylates the Gli transcription factors to target them for proteolytic processing into transcriptional repressor forms (Gli-R). When Hh signaling is activated, the Hh pathway transducer protein Smoothened (Smo) translocates to the primary cilium, and promotes Hh signaling by inhibiting PKA activity, allowing the accumulation of full length Gli proteins, which can act as transcriptional activators (Gli-A) [9,10]. Hh functions as a morphogen, and directs distinct transcriptional outcomes depending on the concentration and time of exposure of responding cells to Hh [80]. The transcriptional outcome is determined by the ratio of Gli-R to Gli-A. In the absence of Hh, PKA activity is high. A partial reduction of PKA activity will reduce the production of Gli-R, and lead to activation of low level target genes, whereas complete inhibition of PKA activity is required for maximal level Hh target gene expression. Thus, the level of PKA activity determines the transcriptional outcome of Hh signaling, and a central aspect of Hh signal transduction is to fine-tune PKA activity to ensure the appropriate response (Figure 3) [9,10,81].

## 5. The Gpr161-cAMP-PKA Signaling Axis in the Primary Cilium

The primary cilium is essential for this response [82]. The core components of the Hh signaling pathway localize to cilia [9,83,84,85], which act as a platform for signal transduction and processing of the Gli transcription factors [4]. Thus, in mouse, loss of primary cilia leads to deregulation of Hh signaling, with a loss of high level target gene expression (Gli-A), and an expansion of low level target gene expression (loss of Gli-R) [4]. Initial work using immunohistochemistry showed that both PKA-C and the regulatory subunit RIIβ localize to the base of the cilium, in the region around the basal body [86,87], suggesting a model where PKA at the base of the cilium may phosphorylate the Gli transcription factors as they exit the cilium [87]. More recently, however, a proteomic analysis of the primary cilium identified the presence of both PKA-C, as well as the two regulatory subunits RIα and RIIβ, within the ciliary shaft [88]. Furthermore, these authors showed that targeting the PKA inhibitory peptide PKI to the primary cilium resulted in loss of processing of Gli, suggesting that PKA can act within the cilium to phosphorylate Gli for proteolytic processing [5,88]. One further interconnection between PKA abundance and Hh signalling involves the Rho GTPase activating protein ARHGAP36 [89].

The direct visualization of cAMP dynamics in the cilium has proved to be technically challenging, and measurements of resting cAMP levels in primary cilia vary [90,91,92,93,94]. Indirect evidence for activation of AC and elevation of cAMP levels in the primary cilium in the absence of Hh ligands is provided by studies on the involvement of GPCRs in the regulation of Hh signaling. The orphan Gαs-coupled receptor Gpr161 has been shown to play a major role in antagonizing Hh signaling [95]. In the absence of Hh ligands, Gpr161 localizes to the primary cilium, where it couples to Gαs to maintain high levels of cAMP and PKA activity. Activation of Hh signaling, as well as the ciliary translocation of Smo, induce the ciliary exit of Gpr161, thus effectively lowering the levels of ciliary cAMP [95]. Loss of Gpr161 results in hyperactivation of Hh signaling, although not to the same extent as a complete loss of PKA activity or a complete loss of Gαs [81,87,95], suggesting that additional GPCRs contribute to the activation of PKA and the maintenance of the basal Hh repression machinery [81]. Consistent with this, additional G protein coupled receptors, including Gpr175, Gpr17, and the chemokine receptor Cxcr4, have been shown to play a role in the regulation of Hh signaling [96,97,98]. However, Gpr161 is unique in this context, as, to date, it is the only GPCR shown to act as an AKAP [69]. The AKAP domain of Gpr161 is selective for type I holoenzymes, and has been shown to recruit cAMP binding RIα subunits to primary cilia of zebrafish embryos [45,69]. One possibility is that Gpr161 acts as an AKAP to recruit the whole type I PKA holoenzymes to the cilium for compartmentalized phosphorylation of the Gli transcription factors. In this scenario, Gpr161 would play a pivotal role in the negative regulation of Hh signaling, by ensuring the correct spatial localization of PKA, and also by regulating the activation of PKA by Gαs-coupling and AC activation. However, type I PKA activities have a very narrow range and are thought to primarily phosphorylate targets that are in very close proximity [11,99].

Interestingly, Gpr161 also contains a PKA consensus phosphorylation site, and has been shown to be a target for PKA phosphorylation [69]. This suggest that Gpr161 itself may be a major target for Gpr161-recruited ciliary type I PKA holoenzymes and may represent a further compartmentalization step within the cilium. The functional significance of PKA phosphorylation of ciliary compartmentalized Gpr161 is not known. However, phosphorylation of these GPCRs has been shown to alter activation state and/or subcellular localization, raising the possibility that PKA phosphorylation regulates ciliary Gpr161 activity and/or localization [69,100]. Alternatively, phosphorylation of Gpr161 may regulate the recruitment of further factors to form a macromolecular AKAP-kinase complex that contributes to regulation and fine-tuning of ciliary signal transmission. A cAMP-sensing Gpr161/PKA type I signalosome could thus be a platform for integrating and orchestrating ciliary cAMP oscillations and PKA activity in the regulation of Hh signaling. A better understanding of the interplay between Gpr161 and PKA in the cilium will require functional characterization of the AKAP and PKA phosphorylation mutants of Gpr161 and structural information about the organization of the Gpr161 signalosome complex.

It is an open question if the cAMP-sensing Gpr161/PKA-type I signalosome is indeed the central platform for integrating and orchestrating ciliary cAMP fluxes, compartmentalized type I PKA activities, Gli processing, and Hh counter-regulation. The next major progress for shedding light on the underlying mechanism of ciliary Gpr161 function will be accomplished upon the identification of Gpr161 ligands. In addition, detailed structural information of the organization of the Gpr161–signalosome complex will be essential for understanding ciliary compartmentalization and its role in the basal repression of Hh signaling [6,101].

Recently, it has been shown that cAMP-induced PKA activities are not just central to Hh regulation. General cAMP fluxes have been linked to regulate cilium formation/removal. This is related to the recruitment of PKA and NIMA related kinase 10 (NEK10), a protein that is essential for ciliogenesis in vertebrates, to the pericentriolar matrix through the pericentriolar material protein 1 (PCM1). Thus, high cAMP levels lead to PKA-mediated phosphorylation of NEK10, which triggers its degradation and ultimately results in cilia resorption [102]. Interestingly, CHIP, the same ubiquitin ligase that marks NEK10 for degradation, is also involved during the regulation of PKAc levels in a negative feedback mechanism that acts to restrict PKA activation [103]. Similarly ARHGAP36 is also involved in limiting PKAc activity and abundance, which in turn regulates the depression of the Hh signaling pathway [89].

Cilia host a collection of GPCRs that are coupled to Gαs or Gαi. The following reviews summarize the identity of ciliary localized GPCRs [3,6,7] and one recent paper indicates the coupling of a selection of ciliary GPCRs [91]. Besides ligand availability, constitutive receptor activities, receptor abundance, and receptor desensitization, it is the import/export of the inventory of the ciliary cAMP platforms that affects Hh antagonizing effects of cAMP. AC3, AC5, and AC6 have been shown to be part of ciliary cAMP production machineries in vertebrates [5] and in response to regulatory input signals from the GPCRs. In this context, it is of major interest that, besides other signaling molecules, PDEs are also central components of AKAP-PKA complexes. In addition to AC activation, different PDE isoforms control cAMP oscillations in a nanodomain-specific manner in the sensory primary cilium compartment. Here, we would like to summarize the available evidence on PDE involvement. PDE1C has been identified in cilia of olfactory sensory neurons [104]. Moreover, PDE2 and guanylyl cyclases (GC) are also expressed in these cilia, where odorant GC-cGMP activation stimulates PDE2 to hydrolyze cAMP/cGMP [105,106]. Another PDE that is linked to cilium functions is the cGMP-dependent PDE6d, a prenyl-binding protein and chaperone of prenylated small G proteins, which is involved in the import of lipidated proteins into the primary cilium [107].

Further, a macromolecular protein complex comprising the ciliary calcium channel polycystin-2, AC5/6, PKA, PDE4c, and the A-kinase anchoring protein AKAP150 has been identified within the primary cilia of renal epithelial cells [108]. Polycystic kidney disease (PKD) is a genetic disorder characterized by the formation of cysts within the kidney tubules, which is caused by mutations that affect the molecular architecture of primary cilia [109]. It has been shown that PKD causing mutations of polycystin-2 led to dysregulated cAMP production within primary cilia. Therefore, cAMP signaling has been suggested to be a promising target for treatment strategies for PKD [108,110,111]. These results highlight that ciliary cAMP levels need to be tightly controlled to ensure healthy physiology, not only during Hh signaling.

## 6. Perturbations of the cAMP and Hh Signaling at the Primary Cilium

Hh regulates developmental and physiological cell/tissue states by supervising cell growth, survival, fate, and patterning [85,112]. In the Hh off-state, ciliary cAMP levels are elevated, which triggers the PKA-mediated proteolytic processing into Gli-R forms, which inhibits Hh target gene expression. Hh ligand availability advances the on-state. This leads to the ciliary accumulation of the central transducing component of the Hh signaling, which is the activated Smo receptor. The integrity of the Hh initiated signaling pathway is strongly associated with ciliary malfunctions. Diseases with cilium dysfunctions are termed ciliopathies. They are linked to mutations of a rising number of ciliopathy-associated genes [109]. A collection of developmental abnormalities is associated with syndromic ciliopathies. Two examples of compromised Hh signaling are polydactyly in the Bardet-Biedl syndrome and neural defects of the Meckel syndrome [109]. Many ciliopathies are caused by a disruption of the ciliogenic program. This may involve perturbations of gene expression, early ciliogenesis and the intraflagellar transport machinery, the formation of the transition zone, or the ciliary translocation of signaling components [10,109]. The central role of Smo as the key Hh signal transducer has focused attention on Smo in the work to develop therapeutic interventions. Especially in basal cell carcinoma or medulloblastoma, distinct somatic mutations promote the Hh pathway. Besides loss of function mutations in the negative Hh regulators Ptch and Sufu, activating mutations or amplifications in Smo and Gli protein variants also promote Hh signaling [113]. At this point, we would like to refer to reviews that precisely describe the collection of small molecule inhibitors and activators of the different components of the Hh signaling pathway [113,114,115]. A considerable number of Smo antagonists are either in clinical or preclinical trials. Two of the recently approved Smo inhibitors are Vismodegib (GDC-0449) and Sonidegib (Erismodegib). Overall, Smo inhibitors show promising antitumor effects against a variety of tumor types. However, there is increasing evidence that mutations in the pathway lead to acquired resistance mechanism [116].

The central role of PKA activity in the basal repression of Hh signaling is well documented. However, there is some ambiguity about how activation of Smo relieves the inhibitory effect on Gli-dependent transcriptional activation [9]. The underlying cell type-specific mechanistic details may depend on the spatiotemporal interplay, and the composition of several factors in and outside of the primary cilium. This also explains why we would rather pinpoint alternative or combinatory targeting strategies that should boost ciliary cAMP levels and PKA activation. There are pharmaceutical targets at different stages of the ciliary signaling axis, which involves diverse druggable components that influence cAMP dynamics. First, one explicit feature of the cilium is the high concentration of GPCRs that are accessible drug targets for intervening with the G protein coupling and AC activation [15,20]. Comprehensive lists of ciliary GPCRs have been published that list either agonist or antagonists of these receptors that promote AC activation and ciliary cAMP production [3,6,7,90,91,95,100]. Second, recent advances in the development of isoform-selective AC inhibitors and activators suggest that ACs could serve as useful drug targets [117]. In this context, one interesting question would be how to target selectively cilium specific AC functions. Recent proof-of concept studies presented nanoparticle-based drug delivery approaches that can specifically target primary cilia in cell culture and animal models [118,119]. Third, the sole means of degrading cAMP are PDEs. They are critically involved in setting the subcellular cAMP concentration, and hence the PKA activity profile. Besides the inhibition of isoform-specific and cilium-enriched PDEs, enhancers of catalytic activity have also been successfully designed [30]. Antagonizing cilium-localized PDE functions may be one feasible strategy to elevate ciliary compartmentalized cAMP levels. Fourth, a collection of PKA-specific cAMP analogs are available. This fact needs to be considered for activating ciliary PKA type I or II holoenzymes, which may help to target scaffolding complexes with a precise enzyme composition [120,121]. 

Last, but not least, we envision that one specific GPCR may become an important candidate for antagonizing deregulated Hh signaling functions through cAMP mobilization. Gpr161 activities seem to be one of the long-sought factors for creating spatio-temporal controlled cAMP-gradients in the ciliary Hh signaling pathway [6,122]. Smo activation and ciliary accumulation lead to the clearance of Gpr161 from the cilium. So far, no ligands for Gpr161 have been identified. Indirect evidence suggests that it is coupled to Gαs and subsequent cAMP elevation. We assume that this receptor might become a feasible target for elevating ciliary cAMP levels. Moreover, future research may reveal if the multimeric complex consisting of cAMP-sensing PKA and Gpr161 is the missing link and central ciliary signalosome to integrate and coordinate cAMP levels, compartmentalized PKA activities, Gli processing, and counteracting Smo signaling. Modulating Gpr161 functions using bioactive small molecules offers the unique opportunity to counteract decontrolled Hh signaling by engaging a ciliary and PKA-linked GPCR pathway. In light of a variety of ciliary dysfunctions and the drawbacks of Smo-directed therapies, we envision that implementing Gpr161 agonists into combination therapy-directed studies should improve the efficacy of Hh inhibition.

## Figures and Tables

**Figure 1 cells-09-00118-f001:**
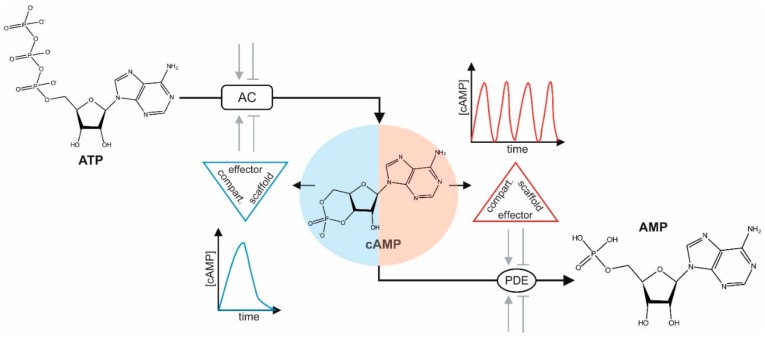
3′,5′-cyclic adenosine monophosphate (cAMP) dynamics. Different means modulate adenylyl cyclases (ACs) either positively or negatively (e.g., extracellular stimuli activate G-protein coupled transmembrane receptors (GPCRs) linked to Gαs or Gαi). Shown are the chemical structures of adenosine triphosphate (ATP), cAMP, and AMP. Spatiotemporal cAMP diffusion is restricted by the coordinated actions and dynamics of protein abundance, localization of receptor-controlled AC activities, phosphodiesterases (PDEs), cAMP-effectors, and physical barriers. Different compartmentalized scaffold–effector complexes (kinase/scaffold/PDE/AC) coordinate frequencies and peaks of cAMP oscillations in space and time.

**Figure 2 cells-09-00118-f002:**
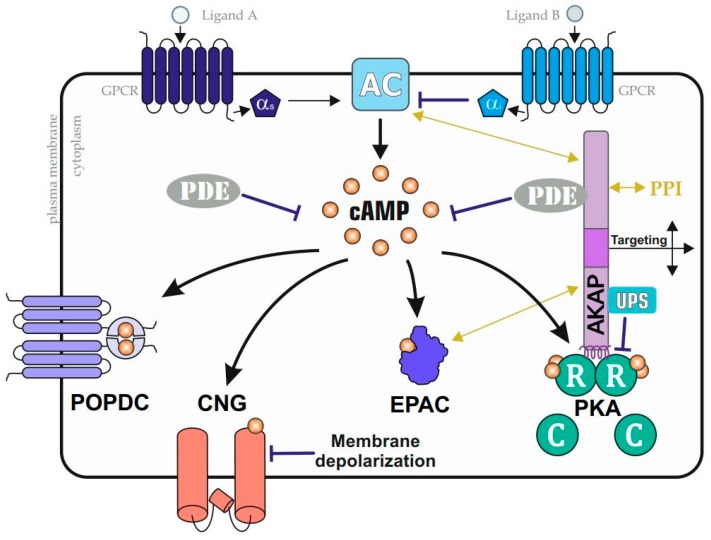
Prominent cAMP–effector pathways. A collection of GPCRs modulates adenylyl cyclase (AC) activities. AC stimulating or inactivating G alpha proteins (Gαs or Gαi, respectively) control cAMP accumulation, which is sensed by a collection of indicated cAMP-binders. In addition to cytoplasmic cAMP-dependent exchange protein (Epac) and protein kinase A (PKA) complexes, membrane-bound cyclic nucleotide-gated ion (CNG) channels and Popeye domain-containing proteins (POPDC) proteins sense cAMP fluxes. PKA is compartmentalized by A kinase anchoring proteins (AKAPs), which coordinate the formation of cell- and context-specific subcellular PKA nanodomains through functional interactions with phosphodiesterases (PDEs); the ubiquitin proteasome system (UPS); and/or through additional PPIs (with, for example, ACs and Epacs), which are AKAP and nanodomain specific (indicated by the yellow arrows).

**Figure 3 cells-09-00118-f003:**
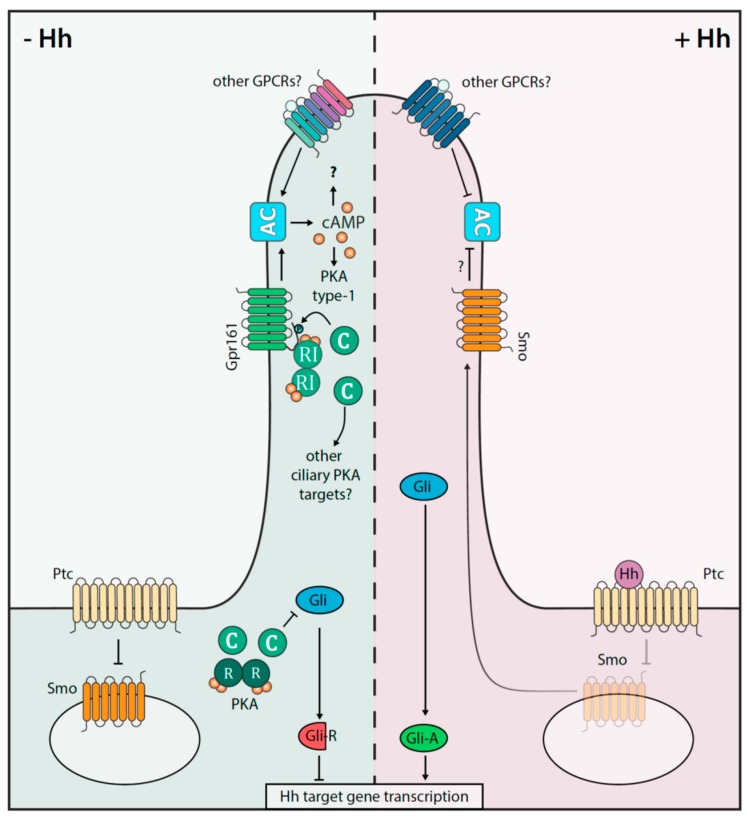
Interplay of the cAMP-PKA axis and Hh signaling in the primary cilium. Shown is a simplified depiction of the Hh on (+Hh) and off (−Hh) states in mammalian cells. Gpr161 and Smoothened (Smo) receptors act as key signal transducer. The involvement of other GPCRs, which impact cAMP and thus Hh signaling, are indicated. In the absence of the Hh ligand, Ptc inhibits Smo signaling. In a PKA dependent-manner, the repressor forms of Gli transcription factors (=Gli-R) are processed. Gli-R prevent the Hh directed gene expression profiles. Upon Hh binding to Ptc, activated Smo is recruited into the cilium, where it counteracts AC activities and Gpr161-PKA signalosomes are exported. The consequence is that the activated transcription factor Gli-A is enriched to promote Hh target gene expression.

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
