# Peer review of "Hedgehog and Gpr161: Regulating cAMP Signaling in the Primary Cilium"

_cells, 2020, doi:10.3390/cells9010118_

Round 1
Reviewer 1 Report
This review offers a general recap of second messenger signaling utility and cAMP mobilization through GPCR signal transduction. The authors then mention the known cAMP binding proteins. From here the article transitions to heavy focus on PKA and its regulation by Gpr161 with implications for Hh signaling in the primary cilium. The case is then made that this signaling axis may represent a therapeutic signaling node in cases of pathological tissue growth (i.e. tumor development). Below are several points of concern.
The manuscript rapidly narrows from a general cAMP discussion to PKA-Gpr161-Hh signaling and as such the title is too general and should be adjusted as to not misinterpret what is being offered in the text. Alternatively, the text could be extended to include more general actions of cAMP in the primary cilium.
The first half of the introduction provides a cryptic overview of cellular signaling systems. This offers a lot of fluff and the manuscript may be strengthened by omission of this test and concisely beginning with sentence that starts “Primary cilia are dynamic organelles that are formed...”
Figure 1 is not referenced in the main text.
Section 3:
The authors seem to downplay the importance of Epac, POPDC, and CNG in their quest to discuss PKA. If this is because Epac, POPDC, and CNG do not have a known role in the primary cilium then this should be stated. Otherwise it may be beneficial to describe what is known about Epac, POPDC, and CNG in the cilia.
Section 4:
The title should be changed to reflect the content of this chapter: PKA and Hh in the primary cilium
The third and fourth sentences should have citations
Reference 86 needs to be corrected
Section 5:
The direct visualization of cAMP in primary cilium has also been observed in PMIDs: 23936473 and 31024067 and these references should be cited as well.
The last paragraph of this section requires improvement. Polycystin-1 also plays a major role in the macromolecular organization of polycystin-2-associated ciliary content. Most ciliary processes are deconstructed in polycystic kidney disease (not just AKAP150). It would be better to cite the original articles in this section rather than the two broad review articles that are currently cited (starting with PMID: 21670265).
Section 6:
This chapter suffers from redundancy. One logical organization may be to merge the first three paragraphs with Section 4 while merging the final paragraph with Section 5.
The reference PMID: 24316073 should be included in the list of ciliary GPCRs implicated in cAMP
The authors mention the possibility of targeting cilia-specific adenylyl cyclase isoforms and this idea may benefit by citing previous successful attempts of delivery content to primary cilia (PMIDs: 30582331, 30860808).
Author Response
General Reviewer 1 Response:
This review offers a general recap of second messenger signaling utility and cAMP mobilization through GPCR signal transduction. The authors then mention the known cAMP binding proteins. From here the article transitions to heavy focus on PKA and its regulation by Gpr161 with implications for Hh signaling in the primary cilium. The case is then made that this signaling axis may represent a therapeutic signaling node in cases of pathological tissue growth (i.e. tumor development). Below are several points of concern.
Referee 1 remarks to the author:
The manuscript rapidly narrows from a general cAMP discussion to PKA-Gpr161-Hh signaling and as such the title is too general and should be adjusted as to not misinterpret what is being offered in the text. Alternatively, the text could be extended to include more general actions of cAMP in the primary cilium.
Author Response: We would like to thank the reviewer for the constructive suggestions. We have changed the title and in the revised manuscript we focus now - as the new title suggests - on how Hh and Gpr161 regulates cAMP.
See page 1 – title and abstract:
Hedgehog and Gpr161: regulating cAMP signaling in the primary cilium.
Abstract: Compartmentalization of diverse types of signaling molecules contributes to the precise coordination of signal propagation. The primary cilium fulfills this function by acting as a spatiotemporally confined sensory signaling platform. For the integrity of ciliary signaling, it is mandatory that the ciliary signaling pathways are constantly attuned by alterations in both, oscillating small molecules and the presence or absence of their sensor/effector proteins. In this context, ciliary G protein-coupled receptor (GPCR) pathways participate in coordinating the mobilization of the diffusible second messenger molecule cAMP. cAMP fluxes in the cilium are primarily sensed by protein kinase A (PKA) complexes, which are essential for the basal repression of Hedgehog (Hh) signaling. Here, we describe the dynamic properties of underlying signaling circuits, as well as strategies for second messenger compartmentalization. As an example, we summarize how receptor-guided cAMP-effector pathways control the off state of Hh signaling. We discuss the evidence that a macromolecular, ciliary-localized signaling complex, composed of the orphan GPCR Gpr161 and type I PKA holoenzymes, is involved in antagonizing Hh functions. Finally, we outline how ciliary cAMP-linked receptor pathways and cAMP-sensing signalosomes may become targets for more efficient combinatory therapy approaches to counteract dysregulation of Hh signaling.
Referee 1 remarks to the author:
The first half of the introduction provides a cryptic overview of cellular signaling systems. This offers a lot of fluff and the manuscript may be strengthened by omission of this test and concisely beginning with sentence that starts “Primary cilia are dynamic organelles that are formed...”
Author Response: We agree. We kicked out the fluff and we present a more concise introduction.
See page 1/2 – introduction:
Compartmentalization is an important aspect of intracellular signaling, both in contributing to the precise coordination of signal propagation, and in ensuring specificity of signaling outcomes [1,2]. Primary cilia are microtubule-based antenna-like structures that extend from the plasma membrane, and represent a highly specialized subcellular compartment/environment for sensing and signaling [3,4]. The composition of the cilium, both in terms of signaling molecules, and also the lipid composition of the ciliary plasma membrane, is strictly regulated, with the transition zone operating to restrict diffusion and control trafficking in and out of the cilium [5]. The last couple of decades have seen an increased focus on, and understanding of, the role that cilia play in regulating signaling. This lead to the identification of several signaling molecules, including G-protein coupled transmembrane receptors (GPCRs) and second messengers, that localize to cilia [3,6,7] .
One central ciliary signaling component is the conserved and ubiquitous second messenger 3', 5'-cyclic adenosine monophosphate (cAMP). First discovered by Sutherland and Rall in 1958, cAMP was proposed to act as a signal effector molecule downstream of hormones like epinephrine, dopamine and prostaglandin [8]. Although this proposal was initially met with skepticism because it was difficult to see how the various actions of distinct hormones could all be mediated by a single effector molecule, we now know that second messengers like cAMP direct different cellular outcomes by interacting with different binding partners in distinct subcellular locations. Through cAMP-dependent protein kinase A (PKA), one of its primary targets, cAMP also plays a pivotal role in the regulation of Hedgehog (Hh) signaling in primary cilia [5,9,10].
Referee 1 remarks to the author:
Figure 1 is not referenced in the main text.
Author Response: Sorry for the mistake. We have changed it.
See page 2:
cAMP mobilizationIntracellular cAMP mobilization is regulated by a family of adenylyl cyclases (ACs) which synthesize cAMP from adenosine triphosphate (ATP) (Figure 1) [12].
Referee 1 remarks to the author:
The authors seem to downplay the importance of Epac, POPDC, and CNG in their quest to discuss PKA. If this is because Epac, POPDC, and CNG do not have a known role in the primary cilium then this should be stated. Otherwise it may be beneficial to describe what is known about Epac, POPDC, and CNG in the cilia.
Author Response: We have changed the text accordingly. See below.
See page 3:
Third, cAMP binds to membrane localized CNGs to regulate their cation gating properties [56]. CNG functions are linked to signal transduction processes in olfactory receptor neurons[57]. In contrast to EPACs and POPDCs which have not been directly linked to ciliary functions CNGs localise to the membranes of specialized cilia of the olfactory sensory neurons and thus are central for the translation of sensory cues into neuronal activity [58].
Referee 1 remarks to the author:
The title should be changed to reflect the content of this chapter: PKA and Hh in the primary cilium. The third and fourth sentences should have citations. Reference 86 needs to be corrected. The direct visualization of cAMP in primary cilium has also been observed in PMIDs: 23936473 and 31024067 and these references should be cited as well.
Author Response: We have changed the title as the reviewer suggested and we have streamlined the manuscript and checked & added the necessary citations. We also list the suggested citations. Thank you for listing them.
See page 1 – title and abstract:
Hedgehog and Gpr161: regulating cAMP signaling in the primary cilium.
Citations:
Marley, A.; Choy, R.W.; von Zastrow, M. Gpr88 reveals a discrete function of primary cilia as selective insulators of gpcr cross-talk. PLoS One 2013, 8, e70857. Sherpa, R.T.; Mohieldin, A.M.; Pala, R.; Wachten, D.; Ostrom, R.S.; Nauli, S.M. Sensory primary cilium is a responsive camp microdomain in renal epithelia. Scientific reports 2019, 9, 6523.Referee 1 remarks to the author:
The last paragraph of this section requires improvement. Polycystin-1 also plays a major role in the macromolecular organization of polycystin-2-associated ciliary content. Most ciliary processes are deconstructed in polycystic kidney disease (not just AKAP150). It would be better to cite the original articles in this section rather than the two broad review articles that are currently cited (starting with PMID: 21670265).
Author Response: We have changed the paragraph. See below.
See page 8
Further, a macromolecular protein complex comprising the ciliary calcium channel polycystin-2, AC5/6, PKA, PDE4c, and the A-kinase anchoring protein AKAP150 has been identified within the primary cilia of renal epithelial cells [107]. Polycystic kidney disease (PKD) is a genetic disorder characterized by the formation of cysts within the kidney tubules, which is caused by mutations that affect the molecular architecture of primary cilia [108]. It has been shown that PKD causing mutations of polycystin-2 lead to dysregulated cAMP production within primary cilia. Therefore, cAMP signaling has been suggested to be a promising target for treatment strategies for PKD [107,109,110]. These results highlight that ciliary cAMP levels need to be tightly controlled to ensure healthy physiology, not only during Hh signaling.
Citations:
Choi, Y.H.; Suzuki, A.; Hajarnis, S.; Ma, Z.; Chapin, H.C.; Caplan, M.J.; Pontoglio, M.; Somlo, S.; Igarashi, P. Polycystin-2 and phosphodiesterase 4c are components of a ciliary a-kinase anchoring protein complex that is disrupted in cystic kidney diseases. Proceedings of the National Academy of Sciences of the United States of America 2011, 108, 10679-10684. Reiter, J.F.; Leroux, M.R. Genes and molecular pathways underpinning ciliopathies. Nat Rev Mol Cell Biol 2017, 18, 533-547. Calvet, J.P. The role of calcium and cyclic amp in pkd. In Polycystic kidney disease, Li, X., Ed. Brisbane (AU), 2015. Padovano, V.; Podrini, C.; Boletta, A.; Caplan, M.J. Metabolism and mitochondria in polycystic kidney disease research and therapy. Nat Rev Nephrol 2018, 14, 678-687.Referee 1 remarks to the author:
This chapter suffers from redundancy. One logical organization may be to merge the first three paragraphs with Section 4 while merging the final paragraph with Section 5.
The reference PMID: 24316073 should be included in the list of ciliary GPCRs implicated in cAMP. The authors mention the possibility of targeting cilia-specific adenylyl cyclase isoforms and this idea may benefit by citing previous successful attempts of delivery content to primary cilia (PMIDs: 30582331, 30860808).
Author Response: We have included the suggested citations. Due to our changes in the introduction we believe that we have reduced the redundancy. We further streamlined the text. As the new title suggests - we focus on Hh - Gpr161 – cAMP and therefore we would be happy to keep the ductus.
Citations:
Loktev, A.V.; Jackson, P.K. Neuropeptide y family receptors traffic via the bardet-biedl syndrome pathway to signal in neuronal primary cilia. Cell reports 2013, 5, 1316-1329. Pala, R.; Mohieldin, A.M.; Shamloo, K.; Sherpa, R.T.; Kathem, S.H.; Zhou, J.; Luan, Z.; Zheng, J.G.; Ahsan, A.; Nauli, S.M. Personalized nanotherapy by specifically targeting cell organelles to improve vascular hypertension. Nano Lett 2019, 19, 904-914. Pala, R.; Mohieldin, A.M.; Sherpa, R.T.; Kathem, S.H.; Shamloo, K.; Luan, Z.; Zhou, J.; Zheng, J.G.; Ahsan, A.; Nauli, S.M. Ciliotherapy: Remote control of primary cilia movement and function by magnetic nanoparticles. ACS Nano 2019, 13, 3555-3572.

Reviewer 2 Report
The authors write an interesting and timely review of cAMP signaling within the cilium. Strengths of the review include a comprehensive discussion of roles for Gpr161 and the general overview of cAMP signaling. However, there is little discussion of roles for other GPCRs and only a passing mention of AKAP-driven complexes within the cilia. These and minor edits are discussed below.
1) The authors need to update references/reviews for PDE’s and AC’s (and other items). For example, the language discussing PDEs as functional barriers for cAMP (bottom of page 2) is quite controversial these days. It might simply be safer to discuss breakdown of cAMP. References throughout are quite old (not always bad, but…). For example, top of page 5 refers to “recent reviews”; the newest of which is 2010.
2) The model Fig 1 suggests that AC and PDE are outside the effector-scaffolded compartment. Yet numerous papers show direct AC binding to AKAPs and its requirement for localized signaling, including AKAP150 that resides in the cilia. Fig 2 also ignores the fact that AC and EPAC have both been shown to bind a subset of AKAPs. Given the last statement on the bottom of page 6 (discussing the narrow range of type 1 PKA), a role for AKAP-confined complexes of AC/PDE/RII/effectors may be an important part of signal propagation within the cilia.
3) Gpr161 is clearly a fascinating GPCR, but it is disappointing that a more integrated model is not presented. As discussed above, there is little discussion of AKAPs or other GPCRs and not much mention of the fact that there are different models of cAMP levels in cilia, with recent work (2019 PNAS) that suggests there are little differences between cAMP levels in cilia and cytosol while others suggest high cilia cAMP or sustained (Sherpa RT et al 2019).
4) Note, page 8 – wrong references for polycystic kidney diseases (not “51, 52”) – CHoi YH PNAS 2011 and ?
The authors need help with punctuation for the abstract.It's a hard review to read because it takes a while to get to the point of cilia signaling and the introduction is somewhat out of date, particularly with respect to anchoring of PDEs and ACs to AKAPs. There are also several wrong citations.The authors focus on Gpr161 but pretty much ignore all other GPCRs. It would be nice if a more unified model is presented.
Author Response
General Reviewer 2 Response:
The authors write an interesting and timely review of cAMP signaling within the cilium. Strengths of the review include a comprehensive discussion of roles for Gpr161 and the general overview of cAMP signaling. However, there is little discussion of roles for other GPCRs and only a passing mention of AKAP-driven complexes within the cilia. These and minor edits are discussed below.
Referee 2 remarks to the author:
1) The authors need to update references/reviews for PDE’s and AC’s (and other items). For example, the language discussing PDEs as functional barriers for cAMP (bottom of page 2) is quite controversial these days. It might simply be safer to discuss breakdown of cAMP. References throughout are quite old (not always bad, but…). For example, top of page 5 refers to “recent reviews”; the newest of which is 2010.
Author Response: We followed the advice of the reviewer and changed the statement about PDE functions. Further, we have added recent publications & reviews about.
See page2
Another way to limit the diffusion or abundance of cAMP, is achieved by the presence and activation of phosphodiesterases (PDEs) [26-31]. Different PDEs mediate the breakdown of cAMP by hydrolyzing it to AMP [32]. By using visualization technologies the existence of cell and context specific cAMP nanodomains has been demonstrated [33-41].
Referee 2 remarks to the author:
2) The model Fig 1 suggests that AC and PDE are outside the effector-scaffolded compartment. Yet numerous papers show direct AC binding to AKAPs and its requirement for localized signaling, including AKAP150 that resides in the cilia. Fig 2 also ignores the fact that AC and EPAC have both been shown to bind a subset of AKAPs. Given the last statement on the bottom of page 6 (discussing the narrow range of type 1 PKA), a role for AKAP-confined complexes of AC/PDE/RII/effectors may be an important part of signal propagation within the cilia.
Author Response: Thank you for this comment. We have changed text and figure to highlight that indeed AKAPs may also functionally interact with either ACs and Epac as well. We have changed the text accordingly.
See page 4
Figure 2. Prominent cAMP-effector pathways. A collection of GPCRs modulates adenylyl cyclase (AC) activities. AC stimulating or inactivating G alpha proteins (Gas or Gai respectively) control cAMP accumulation which is sensed by a collection of indicated cAMP-binders. In addition to cytoplasmic Epac and PKA complexes membrane-bound CNG channels and POPDC proteins sense cAMP fluxes. PKA is compartmentalized by A kinase anchor proteins (AKAPs) which coordinate the formation of cell and context-specific subcellular PKA nanodomains through functional interactions with phosphodiesterases (PDE), the ubiquitin proteasome system (UPS) and/or through additional PPIs (with e.g. ACs and Epacs) which are AKAP and nanodomain specific (indicated by the yellow arrows).
3) Gpr161 is clearly a fascinating GPCR, but it is disappointing that a more integrated model is not presented. As discussed above, there is little discussion of AKAPs or other GPCRs and not much mention of the fact that there are different models of cAMP levels in cilia, with recent work (2019 PNAS) that suggests there are little differences between cAMP levels in cilia and cytosol while others suggest high cilia cAMP or sustained (Sherpa RT et al 2019).
Author Response: We would like to thank the reviewer for this suggestion. Actually we have changed now the title and in the revised manuscript we focus now on how Hh and Gpr161 regulates cAMP. However, we underline in figure legend and text that other GPCR interactions are relevant as well by citing the suggested publications.
See page 1 – title and abstract:
Hedgehog and Gpr161: regulating cAMP signaling in the primary cilium.
See page 9:
First, one explicit feature of the cilium is the high concentration of GPCRs which are accessible drug targets for intervening with the G protein coupling and AC activation [15,20]. Comprehensive lists of ciliary GPCRs have been published which list either agonist or antagonists of these receptors that promote AC activation and ciliary cAMP production [3,6,7,89,90,94,99][116].
4) Note, page 8 – wrong references for polycystic kidney diseases (not “51, 52”) – CHoi YH PNAS 2011 and ?
Author Response: We have edited the citation list. Thank you for bringing it up.
Citation:
Choi, Y.H.; Suzuki, A.; Hajarnis, S.; Ma, Z.; Chapin, H.C.; Caplan, M.J.; Pontoglio, M.; Somlo, S.; Igarashi, P. Polycystin-2 and phosphodiesterase 4c are components of a ciliary a-kinase anchoring protein complex that is disrupted in cystic kidney diseases. Proceedings of the National Academy of Sciences of the United States of America 2011, 108, 10679-10684.The authors need help with punctuation for the abstract.It's a hard review to read because it takes a while to get to the point of cilia signaling and the introduction is somewhat out of date, particularly with respect to anchoring of PDEs and ACs to AKAPs. There are also several wrong citations.The authors focus on Gpr161 but pretty much ignore all other GPCRs. It would be nice if a more unified model is presented.
Author Response: According to the feedback of all three reviewer we streamlined the review more into the direction of how the two receptor pathways (Hh & Gpr161) functionally interact and impact cAMP signaling. We list other GPCRs which are involved (+ citations and reorganization of figure 6 figure legend). In addition we present a more concise & up-to date introduction. We reorganized the citations.

Reviewer 3 Report
Cellular cAMP signalling is restricted in time and space. This is achieved by a compartmentalised organisation maintained by scaffolding proteins, which bring together various types of proteins involved in cAMP signalling. In the primary cilium Protein Kinase A mediated cAMP signalling modulates hedgehog signalling.
Here Tschaikner et al. review first cAMP signalling and compartmentalised signalling in general and then discuss cAMP mediated signalling in the primary cilium.
Specific points:
The nucleotides shown in figure 1 should be presented with correct stereochemistry. For example, the ATP molecule seems to contain D-Lyxose and not D-Ribose. First paragraph on page 6. Additional explanation and discussion is required. Which proteins is PCM1 recruiting? Why are NEK10 and PKA considered counteracting kinases? What are the targets of the ubiquitin ligase CHIP and why is this relevant for cilium resorption? Is there a link between cilium resorption and hedgehog signalling? First paragraph on page 9. The discussion on Ptch and Sufu requires some more introduction of these proteins.Author Response
General Reviewer 3 Response:
Cellular cAMP signalling is restricted in time and space. This is achieved by a compartmentalised organisation maintained by scaffolding proteins, which bring together various types of proteins involved in cAMP signalling. In the primary cilium Protein Kinase A mediated cAMP signalling modulates hedgehog signalling. Here Tschaikner et al. review first cAMP signalling and compartmentalised signalling in general and then discuss cAMP mediated signalling in the primary cilium.
Referee 3 remarks to the author:
The nucleotides shown in figure 1 should be presented with correct stereochemistry. For example, the ATP molecule seems to contain D-Lyxose and not D-Ribose.
Author Response: We changed Figure 1. Thank you for bringing it up.
See page 3:
Figure 1. cAMP dynamics. Different means modulate adenylyl cyclases (AC) either positively or negatively (e.g. extracellular stimuli activate GPCRs linked to Gas or Gai). Shown are the chemical structures of ATP, cAMP, and AMP. Spatiotemporal cAMP diffusion is restricted by the coordinated actions and dynamics of protein abundance, localization of receptor-controlled AC activities, phosphodiesterases (PDEs), cAMP-effectors, and physical barriers. Different compartmentalized scaffold-effector complexes (kinase:scaffold:PDE:AC) coordinate frequencies and peaks of cAMP oscillations in space and time.
See page 2:
Intracellular cAMP mobilization is regulated by a family of adenylyl cyclases (ACs) which synthesize cAMP from adenosine triphosphate (ATP) (Figure 1) [12].
First paragraph on page 6. Additional explanation and discussion is required. Which proteins is PCM1 recruiting? Why are NEK10 and PKA considered counteracting kinases? What are the targets of the ubiquitin ligase CHIP and why is this relevant for cilium resorption? Is there a link between cilium resorption and hedgehog signalling?
Author Response: We have streamlined this paragraph. See below:
See page 8:
Recently, it has been shown that cAMP induced PKA activities are not just central to Hh regulation. General cAMP fluxes have been linked to regulate cilium formation/removal. This is related to the recruitment of PKA and NIMA related kinase 10 (NEK10), a protein which is essential for ciliogenesis in vertebrates, to the pericentriolar matrix through the pericentriolar material protein 1 (PCM1). Thus, high cAMP levels lead to PKA mediated phosphorylation of NEK10 which triggers its degradation and ultimately results in cilia resorption [101]. Interestingly, CHIP, the same ubiquitin ligase that marks NEK10 for degradation, is also involved during the regulation of PKAc levels in a negative feedback mechanism that acts to restrict PKA activation [102]. Similarly ARHGAP36 is also involved in limiting PKAc activity and abundance which in turn regulates the derepression of the Hh signaling pathway [88].
First paragraph on page 9. The discussion on Ptch and Sufu requires some more introduction of these proteins.
Author Response: The new focus of the review is the impact of Hh and Gpr161 on cAMP mobilization. We now highlight reviews which focus on the function of Ptch and Sufu. See below:
Citations:
Kong, J.H.; Siebold, C.; Rohatgi, R. Biochemical mechanisms of vertebrate hedgehog signaling. Development 2019, 146. Briscoe, J.; Therond, P.P. The mechanisms of hedgehog signalling and its roles in development and disease. Nat Rev Mol Cell Biol 2013, 14, 416-429.

Round 2
Reviewer 1 Report
The authors response to the comments nicely and the current manuscript version is much improved.
Reviewer 3 Report
This reviewer has no further comments.